# High prevalence of unawareness of HCV infection status among both HCV-seronegative and seropositive people living with human immunodeficiency virus in Taiwan

Chun-Yuan Lee[1,2,3], Pei-Hua Wu[4,5], Meng-Wei Lu[3], Tun-Chieh Chen[3,6], Po-Liang Lu[3,4,7,8]*

1 Department of Internal Medicine, Kaohsiung Municipal Siaogang Hospital, Kaohsiung, Taiwan, 2 Graduate Institute of Medicine, Kaohsiung Medical University, Kaohsiung, Taiwan, 3 Department of Medicine, College of Medicine, Kaohsiung Medical University, Kaohsiung, Taiwan, 4 Division of Infectious Diseases, Department of Internal Medicine, Kaohsiung Medical University Hospital, Kaohsiung, Taiwan, 5 Department of Public Health, College of Health Science, Kaohsiung Medical University, Kaohsiung, Taiwan, 6 Infection Control Office, Kaohsiung Municipal Ta-Tung Hospital, Kaohsiung, Taiwan, 7 Department of Laboratory Medicine, Kaohsiung Medical University Hospital, Kaohsiung, Taiwan, 8 Liquid Biopsy and Cohort Research Center, Kaohsiung Medical University, Kaohsiung, Taiwan

* d830166@gmail.com

**Data Availability Statement:** All relevant data are within the paper and its Supporting Information files.

## Abstract

### Objectives

HCV infection status awareness is crucial in the HCV care continuum for both HCV-seropositive (HCV-positive status awareness) and seronegative (HCV-negative status awareness) populations. However, trends in the unawareness of HCV infection status (UoHCV) remain unknown in HIV-positive patients. This study investigated UoHCV prevalence, the associated factors of UoHCV, and its association with HCV-related knowledge in HIV-positive patients.

### Methods

For this cross-sectional, multicenter, questionnaire-based study, 844 HIV-infected participants were recruited from three hospitals in Taiwan from June 2018 to March 2020. Participants were grouped by HCV serostatus (HCV-seronegative [n = 734] and HCV-seropositive [n = 110]) and categorized by their HIV diagnosis date (before 2008, 2008–2013, and 2014–2020). Exploratory factor analysis was used to categorize the 15 items of HCV-related knowledge into three domains: route of HCV transmission, HCV course and complications, and HCV treatment.

### Results

The prevalence of UoHCV was 58.7%–62.6% and 15.1%–31.3% in the HCV-seronegative and HCV-seropositive groups, respectively, across 3 periods. More participants with

**Funding:** Chun-Yuan Lee received grants from the Ministry of Science and Technology, Taiwan, R.O. C. under grant no. MOST 108-2314-B-037-050 and Kaohsiung Municipal Siaogang Hospital, Taiwan, R.O.C. under grant no.H-108-006 and Kaohsiung Medical University Research Center, Taiwan, R.O. C. under grant no. KMU-TC109B05.

**Competing interests:** The authors have declared that no competing interests exist.

UoHCV believed that HCV infection was only contracted by intravenous injection. In the HCV-seropositive group, participants with UoHCV were more likely to have HIV diagnosis before 2008 (vs. 2014–2020), be men who have sex with men (vs. people who inject drugs), and have hepatitis A virus seronegativity. In the HCV-seronegative group, participants with UoHCV were more likely to have a recent history of sexually transmitted diseases, but had a lower education level, had received less information on HCV infection from clinicians, and were less likely to have heard of HCV infection prior to the research. UoHCV was associated with lower scores for three domains of HCV-related knowledge in both groups.

## Conclusions

The negative association of UoHCV with HCV-related knowledge suggests that strategies targeting patients according to their HCV serostatus should be implemented to reduce UoHCV and eradicate HCV infection among HIV-positive patients.

## Introduction

The global seroprevalence of the hepatitis C virus (HCV) is approximately 2.5% [1]. Although treatment with direct-acting antivirals (DAAs) can lead to the elimination of HCV viremia and a curative outcome in more than 90% of patients with chronic HCV infection [2], several barriers to eradicating HCV infection still exist, including the high costs of drugs [3], frequent loss to follow-up after diagnosis [4], high rate of early HCV reinfection among patients who have recently received drug injections [5], and ongoing high-risk behaviors associated with HCV infection (even after clearance of HCV infection) [6]. The identification of undiagnosed patients, timely provision of DAAs to HCV-seropositive populations [7, 8], and the prevention of transmission among at-risk HCV-seronegative populations must be prioritized to eliminate HCV infection [7].

Awareness of HCV infection status is crucial for both HCV- seropositive and HCV-sero-negative populations. Although the short- and long-term impacts of HCV-positive status awareness among HCV-seropositive patients on their risk behavior remain matters of debate [9–12], HCV-positive status awareness is essential in the HCV care continuum in terms of treatment eligibility and taking medical advice on viral transmission [12, 13]. However, unawareness of HCV infection status (UoHCV) remains common in the HCV-seropositive population, with prevalence rates of 20.5% in Italy [14], 16.6%–35.1% in Taiwan [15–17], and 14%–51% in the United States [18, 19]. In the HCV-seronegative population at risk of contracting HCV, people who inject drugs (PWID) may engage in high-risk behaviors (e.g., sharing a syringe or injecting themselves with drugs) less frequently if they are aware of their HCV infection status (i.e., HCV-negative status awareness) [9, 12]. However, a nationwide screening program in Taiwan revealed a 33% prevalence of UoHCV in the HCV-seronegative population [16].

People living with human immunodeficiency virus (HIV) infection (PLWH) are at risk of HCV infection because the transmission routes of HCV infection, such as unprotected sex and drug injection, are similar to those of HIV infection [6, 20, 21]. Moreover, individuals coinfected with HCV and HIV are less likely to seek HCV care [22–24], which may contribute to a significantly decreased quality of life and quicker progression of liver disease, especially in those who are homeless or marginally housed [25]. Additionally, patients with HCV/HIV

coinfection have higher rates of death and disease progression, including the progression of histological fibrosis/cirrhosis and decompensated liver disease, than do patients with HCV monoinfection [26]. Therefore, HCV screening, treatment, and prevention strategies should be strictly implemented among PLWH [27]. However, although strategies have been implemented worldwide to combat the spread of HCV, no study has explored the prevalence or associated factors of UoHCV among PLWH [11]. Serostatus awareness facilitates the next step in the continuum of HCV care, namely providing affected patients with access to health care, relevant consultation, and potential treatment, which are necessary to eradicate HCV. Furthermore, knowledge regarding HCV infection is essential for the further utilization of HCV treatment [28, 29]. In one study, the majority of participants had limited knowledge regarding the complications of chronic HCV infection despite being aware of the high prevalence of HCV among men who have sex with men (MSM) [30]. However, little is known regarding the association of UoHCV with knowledge of HCV among PLWH. We hypothesized that UoHCV is negatively associated with an individual's HCV-related knowledge, regardless of their HCV serostatus.

We conducted a cross-sectional questionnaire-based study at three HIV referral centers in Taiwan from June 2018 to March 2020. We evaluated the prevalence of UoHCV, explored the determinants of UoHCV, and evaluated the associations of UoHCV with different domains of HCV-related knowledge (i.e., route of HCV transmission, HCV course and complications, and HCV treatment) among a sample of PLWH stratified by HCV serostatus.

## Materials and methods

### Study design and setting

This cross-sectional, multicenter, questionnaire-based study was conducted from June 1, 2018, to March 31, 2020, at Kaohsiung Municipal Siaogang Hospital and Kaohsiung Municipal Ta-Tung Hospital, which are regional hospitals in southern Taiwan, and at Kaohsiung Medical University Hospital, which is the largest referral center for PLWH in southern Taiwan. The HCV seropositivity in southern Taiwan is 8.6% [31], which is higher than that in northern Taiwan (1.2%–2.7%) [32].

### Development of the study questionnaire

A questionnaire was designed to investigate participants' awareness of HCV infection status, knowledge of different aspects of the disease, perceived risk of HCV infection, and assessment of potential exposure to HCV.

An expert group comprising an HIV case manager, HIV specialists, hepatologists, and researchers developed the preliminary questionnaire used in this study. The preliminary questionnaire was then modified based on feedback from 30 PLWH after they had completed a pretest. The questionnaire was tested again with 20 PLWH and further modified. Because of the lack of a standardized scoring system for evaluating different categories of HCV-related knowledge, the questionnaire items were modified after a review of pertinent studies [28–30, 33]. The section on the perceived risk of HCV infection was also modified [34].

The final questionnaire comprised variables in the following five categories: sociodemographic characteristics, awareness of HCV infection status, knowledge of HCV infection, perceived risk of HCV infection, and assessment of potential exposure to HCV (S1 File). Participants were instructed to answer the 15 HCV-related knowledge items by providing one of the following responses: "yes," "no," or "I do not know" [35]. One point was awarded for each correct response, and no point was awarded for incorrect or "I do not know" responses. Therefore, the mean scale scores ranged from 0 to 1, with higher scores indicating greater

HCV-related knowledge. Variables correlated with the respondents' perceived risk of HCV infection were measured on a 5-point Likert scale (strongly disagree, disagree, neither agree nor disagree, agree, and strongly agree).

## Participants and study procedure

Two trained investigators screened PLWH by reviewing their available medical records at the participating hospitals for the period from January 1, 2000, to March 31, 2020. Patients who were less than 20 years old during the screening period (June 1, 2018, to March 31, 2020), had not undergone HCV antibody tests within 1 year before enrollment, or were lost to follow-up during the screening period were excluded. The participants completed the questionnaires on Google Forms.

The participants were classified into one of two groups according to their HCV serostatus: a HCV-seronegative group and a HCV-seropositive group. They were then stratified according to their awareness of their HCV infection status. Finally, each of the two groups was further divided into two subgroups: the unawareness/HCV-seronegative (subgroup 1), awareness/ HCV-seronegative (subgroup 2), unawareness/HCV-seropositive (subgroup 3), and awareness/HCV-seropositive (subgroup 4).

The study was approved by the Institutional Review Board of Kaohsiung Medical University Hospital (KMUHIRB-SV(I)-20180024) and adhered to the principles of the Declaration of Helsinki. The investigators obtained signed informed consent forms from all patients before enrollment.

## Definitions

The participants were categorized by three distinct periods based on the calendar year of their first confirmed HIV diagnosis: before 2008 (period 1, before the remission of the HIV epidemic among PWID) [36], 2008–2013 (period 2, remission of the HIV epidemic among PWID but before the introduction of oral DAAs), and 2014–2020 (period 3, after the introduction of oral DAAs).

In this study, awareness of HCV infection status was defined as participants' self-reported recognition of their HCV infection status at the time of enrollment in the study (i.e., HCV-seropositive patients' awareness of their HCV-positive status and HCV-seronegative patients' awareness of their HCV-negative status), whereas UoHCV was defined as participants' self-reported unawareness of their HCV infection status [8, 16].

The behavioral indicators of a high risk of exposure to HCV infection were modified from other studies and included using any intravenous recreational drugs [12], engaging in chemosexual behaviors within the preceding 6 months [37], having a sexual partner within the preceding 6 months (assessment options were no sexual partners, one regular sexual partner, no regular sexual partners/less than five partners, and no regular sexual partner/more than five partners) [37], and engaging in other activities involving sexual contact within the preceding 6 months [37].

## Outcomes of interest

The primary outcome of interest was the prevalence of UoHCV across the three study periods, stratified by the participants' HCV serostatus. Secondary outcomes were factors associated with UoHCV and the associations of UoHCV with the mean scores for three domains of HCV knowledge (route of HCV transmission, HCV course and complications, and HCV treatment) among the participants stratified by their HCV serostatus.

## Statistical analysis

Descriptive analyses were performed on the characteristics of the participants in the four subgroups. The categorical and continuous variables in each subgroup were compared through $\chi^2$ tests (or Fisher's exact tests) and independent $t$ tests, respectively. The prevalence of UoHCV was defined as the proportion of participants with UoHCV at the time of questionnaire completion. The trend analyses of the prevalence of UoHCV stratified by HCV serostatus in the three periods were performed using a Cochran–Armitage trend test with modified ridit scores.

Backward stepwise binary logistic regressions were performed to calculate odds ratios and evaluate associations in the bivariate and multivariable analyses between surveyed factors and UoHCV among all the participants and among those in the two HCV serostatus groups. To simultaneously consider the effects of all variables in the multivariable model, we adopted a backward approach.

To determine the validity of the 15 items used to measure the participants' knowledge regarding HCV infection, an item analysis was performed for the assessment of item discrimination. We also performed exploratory factor analysis by using principal axis factoring with varimax rotation to investigate the structural domain of the 15 items, and three domains were finally categorized: route of HCV transmission (domain 1), HCV course and complications (domain 2), and HCV treatment (domain 3). Cronbach's α was used to measure the internal consistency of the items in each structural domain, where α represented the function of the number of items in a test. Cronbach's α ≥ 0.7 indicates high reliability.

Finally, to determine the association of UoHCV with the means of the total and domain-specific scores of HCV-related knowledge, we employed a multilinear regression model with a backward approach. $\beta$ along with 95% confidence intervals were calculated to estimate the effects of UoHCV and directions of all associations. A backward approach was also adopted to enable the effects of all the variables to be simultaneously considered in the multivariable model.

All tests were two-tailed, and $p < 0.05$ was considered significant. Statistical analyses were performed using SPSS Statistics version 25.0 (IBM, Armonk, NY, USA).

# Results

## Participants

The study flowchart is displayed in Fig 1. Among the 1448 eligible PLWH, 525 were unwilling to participate in the study, and 79 were excluded for incomplete responses to questionnaire items. A total of 844 PLWH were included in the final analysis. They were divided into HCV-seronegative (n = 734) and HCV-seropositive (n = 110) groups. The two groups were further divided into subgroups 1 (unawareness/HCV-seronegative; n = 448), 2 (awareness/HCV-sero-negative; n = 286), 3 (unawareness/HCV-seropositive; n = 22), and 4 (awareness/HCV-sero-positive; n = 88).

## Characteristics of the study participants

Table 1 presents the sociodemographic characteristics, laboratory profiles, perceived risk of HCV infection, and high-risk behaviors for HCV infection of the participants in each subgroup. The mean (standard deviation) age at enrollment among all the participants was 36.6 (±9.8) years, and 98.1% of the participants were men. The routes of HIV transmission included men who have sex with men (MSM) (73.5%), bisexual contact (13.0%), heterosexual contact

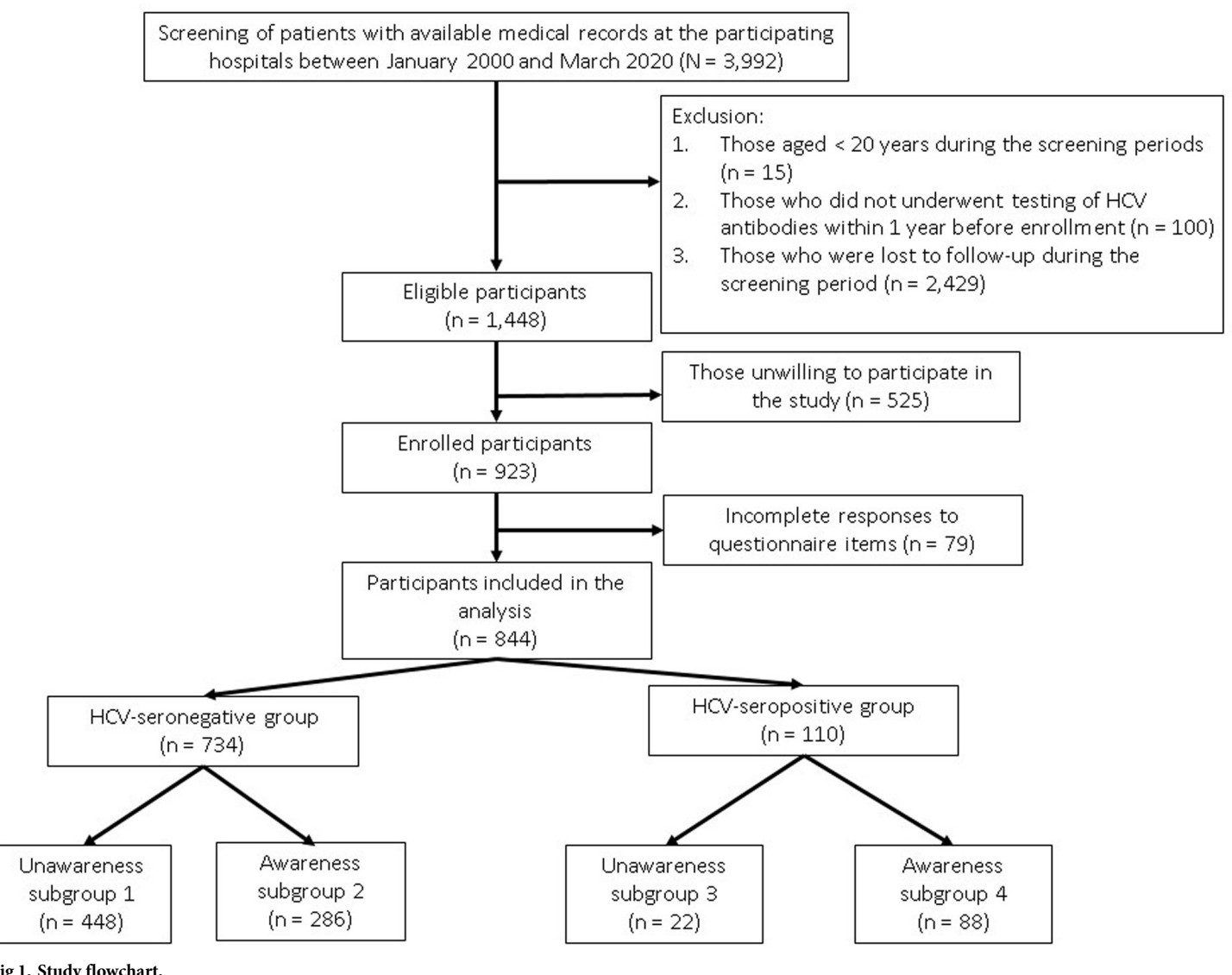

**Fig 1. Study flowchart.**

(8.3%), and drug injection (5.2%). Approximately 21.0%, 30.2%, and 48.8% of the participants were diagnosed as having HIV in periods 1, 2, and 3, respectively.

### Trend analysis of the prevalence of UoHCV across the three periods stratified by participants' HCV serostatus

The prevalence rate of UoHCV was 46.9% in period 1, 54.5% in period 2, and 60.2% in period 3 (p for trend = 0.003; Fig 2). The overall prevalence of UoHCV was 58.7%–62.6% in the HCV-seronegative group (p for trend = 0.497) and 15.1%–31.3% in the HCV-seropositive group (p for trend = 0.086). Participants with a history of treatment for HCV were excluded from the HCV-seropositive group because all of these patients were aware of their HCV infection status and were thus not part of the target population; this exclusion increased the prevalence of UoHCV to 33.8% (ranging from 30.8% in period 1 to 37.0% in period 3, p for trend = 0.632).

**Table 1. Comparison of sociodemographic characteristics of 844 PLWH between those with and without awareness of their HCV infection status, stratified by HCV serostatus.**

| | All N = 844 | HCV-seronegative group | | P | HCV-seropositive group | | P |
|---|---|---|---|---|---|---|---|
| | | Subgroup 1 (Unawareness) | Subgroup 2 (Awareness) | | Subgroup 3 (Unawareness) | Subgroup 4 (Awareness) | |
| | | N = 448 | N = 286 | | N = 22 | N = 88 | |
| **Sociodemographic variables** | | | | | | | |
| • Period of HIV diagnosis, n (%) | | | | 0.621 | | | 0.167 |
| Period 1 (before 2008) | 177 (21.0) | 75 (16.7) | 49 (17.1) | | 8 (36.4) | 45 (51.1) | |
| Period 2 (2008–2013) | 255 (30.2) | 135 (30.1) | 95 (33.2) | | 4 (18.2) | 21 (23.9) | |
| Period 3 (2014–2020) | 412 (48.8) | 238 (53.1) | 142 (49.7) | | 10 (45.5) | 22 (25.0) | |
| • Male gender, n (%) | 828 (98.1) | 443 (98.9) | 281 (98.3) | 0.471 | 21 (95.5) | 83 (94.3) | >0.999 |
| • Age | 36.6 (9.8) | 35.3 (9.8) | 36.6 (9.3) | 0.091 | 38.3 (10.8) | 42.5 (9.4) | 0.70 |
| • Education above college level, n (%) | 452 (53.6) | 226 (50.4) | 193 (67.5) | <0.001 | 7 (31.8) | 26 (29.5) | 0.835 |
| • Employment, n (%) | 690 (81.8) | 360 (80.4) | 250 (87.4) | 0.013 | 15 (68.2) | 65 (73.9) | 0.592 |
| • Marriage, n (%) | 42 (5.0) | 21 (4.7) | 13 (4.5) | 0.389 | 2 (9.1) | 6 (6.8) | 0.403 |
| • HIV diagnosis in Kaoping area, n (%) | 762 (90.3) | 400 (89.3) | 258 (90.2) | 0.689 | 20 (90.9) | 84 (95.5) | 0.345 |
| • HIV-related risk assessment | | | | 0.720 | | | 0.023 |
| MSM | 620 (73.5) | 339 (75.7) | 223 (78.0) | | 15 (68.2) | 43 (48.9) | |
| Heterosexual | 70 (8.3) | 42 (9.4) | 20 (7.0) | | 2 (9.1) | 6 (6.8) | |
| Bisexual | 110 (13.0) | 65 (14.5) | 42 (14.7) | | 2 (9.1) | 1 (1.1) | |
| PWID | 44 (5.2) | 2 (0.4) | 1 (0.3) | | 3 (13.6) | 38 (43.2) | |
| • History of sexually-transmitted diseases within the preceding 6 months, n (%) | 150 (17.8) | 94 (21.0) | 40 (14.0) | 0.017 | 5 (22.7) | 11 (12.5) | 0.307 |
| • Has your doctor ever provide you the information about your HCV infection status before | 369 (43.7) | 95 (21.2) | 193 (67.5) | <0.001 | 10 (45.5) | 71 (80.7) | 0.001 |
| • Have ever heard of HCV | 678 (80.3) | 299 (66.7) | 273 (95.5) | <0.001 | 18 (81.8) | 88 (100.0) | <0.001 |
| **Laboratory data at time of questionnaire, n (%)** | | | | | | | |
| • HAV Ab seropositivity | 538 (64.0) | 276 (61.7) | 187 (65.6) | 0.290 | 11 (50.0) | 64 (74.4) | 0.027 |
| • HBs Ag seropositivity | 84 (10.0) | 45 (10.0) | 26 (9.1) | 0.681 | 2 (9.1) | 11 (12.6) | 0.646 |
| **Perceived risk of HCV infection** | | | | | | | |
| • Only those who inject medication intravenously can get hepatitis C | 2.12 (1.00) | 2.3 (1.02) | 1.86 (0.88) | <0.001 | 2.18 (1.30) | 2.03 (0.99) | 0.558 |
| • The sexual behavior styles that I like put me at risk of hepatitis C infection. | 2.95 (1.09) | 2.99 (0.99) | 2.91 (1.23) | 0.347 | 3.23 (1.11) | 2.81 (1.12) | 0.118 |
| • I am more worried about hepatitis C virus than HIV | 2.90 (1.04) | 2.90 (0.97) | 2.90 (1.16) | 0.952 | 2.64 (1.00) | 3.00 (0.98) | 0.125 |
| **Assessment of potential exposures to HCV within the preceding 6 months** | | | | | | | |
| • Use of intravenous form of recreational drugs, n (%) | | | | 0.126 | | | 0.215 |
| No use | 756 (89.6) | 412 (92.0) | 274 (95.8) | | 18 (81.8) | 52 (59.1) | |
| Yes, less than a year | 35 (4.1) | 23 (5.1) | 5 (1.7) | | 1 (4.5) | 6 (6.8) | |
| Yes, 1–3 years | 20 (2.4) | 10 (2.2) | 5 (1.7) | | 1 (4.5) | 4 (4.5) | |
| Yes, more than 3 years | 33 (3.9) | 3 (0.7) | 2 (0.7) | | 2 (9.1) | 26 (29.5) | |
| • Engagement in chemosexual behaviors within the preceding 6 months, n (%) | 109 (12.9) | 65 (14.5) | 34 (11.9) | 0.311 | 5 (22.7) | 5 (5.7) | 0.013 |
| • Status of having a sexual partner within the preceding 6 months, n (%) | | | | 0.383 | | | 0.832 |
| No sexual partner | 316 (37.4) | 164 (36.6) | 91 (31.8) | | 12 (54.5) | 49 (55.7) | |
| Regular sexual partner | 334 (39.6) | 173 (38.6) | 125 (43.7) | | 7 (31.8) | 29 (33.0) | |
| No regular sexual partners, less than 5 partners | 144 (17.1) | 79 (17.6) | 54 (18.9) | | 3 (13.6) | 8 (9.1) | |
| No regular sexual partner, more than 5 partners | 50 (5.9) | 32 (7.1) | 16 (5.6) | | 0 (0.0) | 2 (2.3) | |
| • Sexual experiences within the preceding 6 months, n (%) | | | | | | | |
| Ever experience sadomasochism | 23 (2.7) | 10 (2.2) | 9 (3.1) | 0.447 | 1 (4.5) | 3 (3.4) | >0.999 |
| Ever experience group sex participation | 82 (9.7) | 50 (11.2) | 27 (9.4) | 0.458 | 1 (4.5) | 4 (4.5) | >0.999 |
| Ever experience Insertive/receptive unprotected anal intercourse | 440 (52.1) | 236 (52.7) | 172 (60.1) | 0.047 | 9 (40.9) | 23 (26.1) | 0.195 |
| Ever experience vaginal sex | 87 (10.3) | 42 (9.4) | 32 (11.2) | 0.426 | 2 (9.1) | 11 (12.5) | >0.999 |

Abbreviations: Ab, antibody; Ag, antigen; HBs, hepatitis B surface; HCV, hepatitis C virus; HIV, human immunodeficiency virus; PWID, people who inject drugs; MSM, men who have sex with men; PLWH, people living with human immunodeficiency virus.

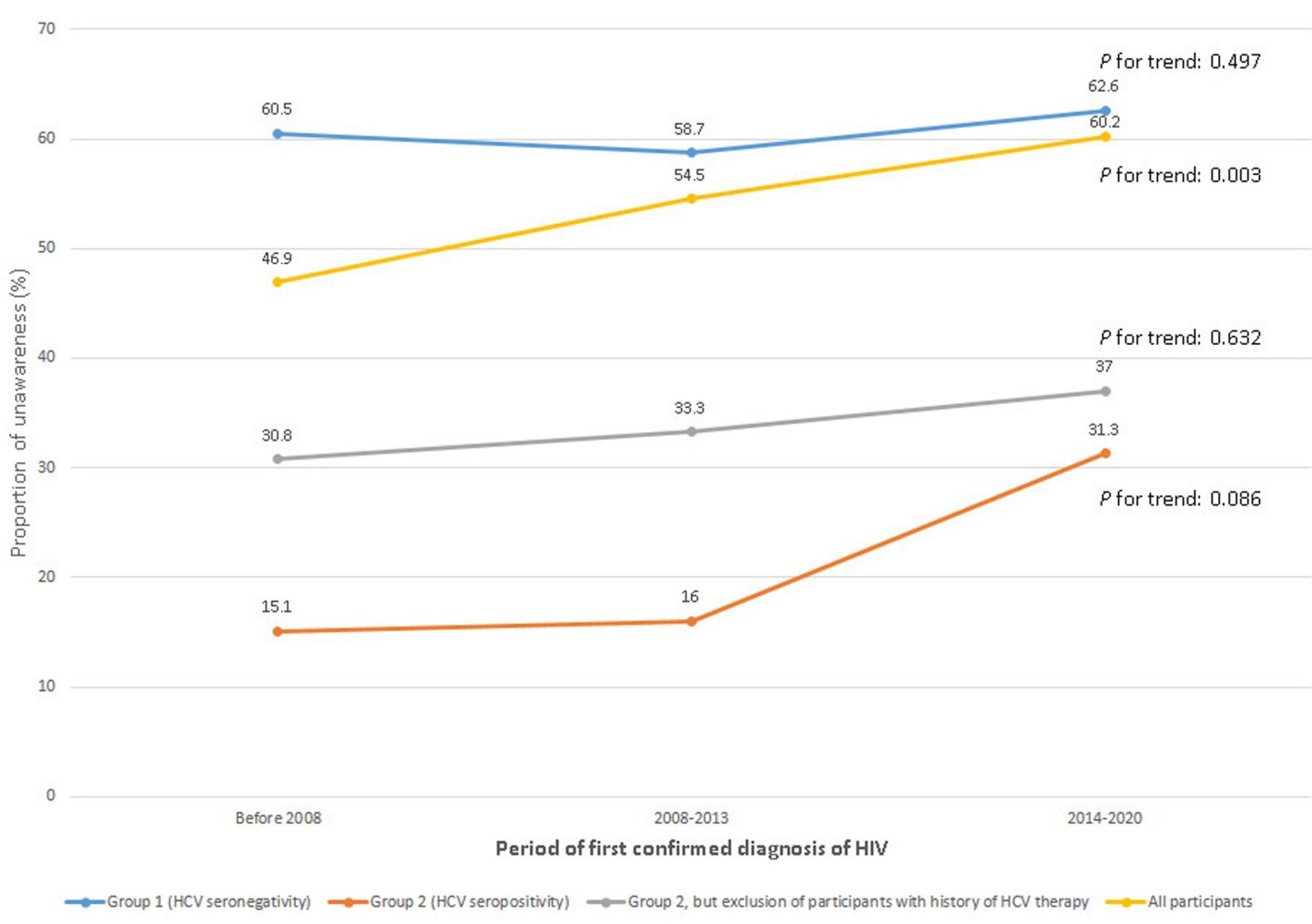

**Fig 2. Trend analyses of prevalence of UoHCV stratified by HCV serostatus in three periods according to the calendar year of first confirmed HIV diagnosis (period 1 [before 2008], period 2 [2008–2013], and period 3 [2014–2020]).** Prevalence of unawareness of HCV infection status across the three periods among all participants and among HCV-seronegative individuals, HCV-seropositive individuals receiving HCV therapy, and HCV-seropositive individuals not receiving HCV therapy. Cochran–Armitage trend test with modified ridit scores was used to analyze the trends in the prevalence of unawareness of HCV infection status for periods 1 to 3. Abbreviations: HCV, hepatitis C virus.

### Factors associated with UoHCV among PLWH stratified by HCV serostatus

After stratification by HCV serostatus, the two groups differed in terms of factors associated with UoHCV in a binary logistic regression (Table 2). In the HCV-seropositive group, the proportion of participants who believed that intravenous injection was a requirement for contracting HCV was greater among individuals with UoHCV than among those who were aware of their HCV status. Furthermore, these individuals with UoHCV were less likely to have received an HIV diagnosis in period 3 (vs. period 1), be PWID (vs. MSM), and have hepatitis A virus (HAV) seropositivity, compared with individuals with awareness of their HCV status. In the HCV-seronegative group, compared with the participants who were aware of their HCV status, those with UoHCV were more likely to have a history of sexually transmitted diseases within the preceding 6 months and believe that intravenous injection was a requirement for contracting HCV. Moreover, they were less likely to have received

**Table 2. Bivariate and multivariable analyses of factors associated with UoHCV among PLWH stratified into two groups: HCV-seronegative group (n = 734) and HCV-seropositive group (n = 110).**

| | | All participants | | | HCV-seronegative group | | HCV-seropositive group | |
|---|---|---|---|---|---|---|---|---|
| | | % of UoHCV | Bivariate analysis, crude OR (95% CI) | Multivariable analysis, adjusted OR (95% CI) | % of UoHCV | Multivariable analysis, adjusted OR (95% CI) | % of UoHCV | Multivariable analysis, adjusted OR (95% CI) |
| **Sociodemographic variables** | | | | | | | | |
| • Period of HIV diagnosis | | | | | | | | |
| | Period 1 (before 2008) | 46.9 | Reference | | 60.5 | | 15.1 | Reference |
| | Period 2 (2008–2013) | 54.5 | 1.357 (0.924–1.994) | | 58.7 | | 16.0 | |
| | Period 3 (2014–2020) | 60.2 | 1.713 (1.201–2.443)** | | 62.6 | | 31.3 | 0.029 (0.001–0.918)* |
| • Gender | | | | | | | | |
| | Female | 37.5 | Reference | | 50.0 | | 16.7 | |
| | Male | 56.0 | 2.125 (0.765–5.900) | | 61.2 | | 20.2 | |
| • Age, per 1-year increase | | N/A | 0.974 (0.961–0.988)*** | | N/A | | N/A | |
| • Education above college level | | | | | | | | |
| | No | 60.5 | Reference | Reference | 70.5 | Reference | 19.5 | |
| | Yes | 51.5 | 0.696 (0.529–0.915)** | 0.481 (0.334–0.692)*** | 53.9 | 0.462 (0.317–0.673)*** | 21.2 | |
| • Employment | | | | | | | | |
| | No | 61.7 | Reference | | 71.0 | | 23.3 | |
| | Yes | 54.3 | 0.739 (0.517–1.057) | | 59.0 | | 18.8 | |
| • Marriage | | | | | | | | |
| | No | 57.2 | Reference | | 61.3 | | 22.2 | |
| | Yes | 39.4 | 0.269 (0.135–0.535)*** | | 57.1 | | 13.8 | |
| • HIV diagnosis in Kaoping area | | | | | | | | |
| | No | 61.0 | Reference | | 63.2 | | 33.3 | |
| | Yes | 55.1 | 0.786 (0.493–1.253) | | 60.8 | | 19.2 | |
| • HIV-related risk assessment | | | | | | | | |
| | MSM | 57.1 | Reference | | 60.3 | | 25.9 | Reference |
| | Heterosexual | 62.9 | 0.204 (0.107–0.391)*** | | 67.7 | | 25.0 | |
| | Bisexual | 60.9 | 0.254 (0.145–0.447)*** | | 60.7 | | 66.7 | |
| | PWID | 11.4 | 0.625 (0.393–0.997)* | | 66.7 | | 7.3 | 0.028 (0.001–0.877)* |
| • History of sexually-transmitted diseases within the preceding 6 months | | | | | | | | |
| | No | 53.5 | Reference | Reference | 59.0 | Reference | 18.1 | |
| | Yes | 66.0 | 1.690 (1.168–2.445)** | 2.012 (1.261–3.212)** | 70.1 | 2.190 (1.349–3.554)** | 31.3 | |
| • Has your doctor ever provide you the information about your HCV infection status before | | | | | | | | |
| | No | 76.8 | Reference | Reference | 79.1 | Reference | 41.4 | |
| | Yes | 28.5 | 0.120 (0.088–0.164)*** | 0.167 (0.118–0.236)*** | 33.0 | 0.154 (0.107–0.222)*** | 12.3 | |
| • Have ever heard of HCV | | | | | | | | |
| | No | 92.2 | Reference | Reference | 92.0 | Reference | 100.0 | |
| | Yes | 46.8 | 0.075 (0.042–0.134)*** | 0.151 (0.081–0.283)*** | 52.3 | 0.158 (0.085–0.297)*** | 17.0 | |
| **Laboratory data at time of questionnaire, n (%)** | | | | | | | | |
| • HAV Ab seropositivity | | | | | | | | |
| | No | 60.3 | Reference | | 63.6 | | 33.3 | Reference |
| | Yes | 3.3 | 0.754 (0.566–1.004) | | 59.6 | | 14.7 | 0.015 (0.001–0.270)** |
| • HBs Ag seropositivity | | | | | | | | |
| | No | 55.8 | Reference | | 60.9 | | 20.8 | |
| | Yes | 56.0 | 1.006 (0.636–1.584) | | 63.4 | | 15.4 | |
| • HCV Ab seropositivity | | | | | | | | |
| | No | 61.0 | Reference | Reference | N/A | | N/A | |
| | Yes | 20.0 | 0.160 (0.098–0.261)*** | 0.428 (0.229–0.800)** | N/A | | N/A | |
| **Perceived risk of HCV infection** | | | | | | | | |

*(Continued)*

**Table 2.** (Continued)

| | | All participants | | | HCV-seronegative group | | HCV-seropositive group | |
| --- | --- | --- | --- | --- | --- | --- | --- | --- |
| | | % of UoHCV | Bivariate analysis, crude OR (95% CI) | Multivariable analysis, adjusted OR (95% CI) | % of UoHCV | Multivariable analysis, adjusted OR (95% CI) | % of UoHCV | Multivariable analysis, adjusted OR (95% CI) |
| • Only those who inject medication intravenously can get hepatitis C | | N/A | 1.527 (1.317–1.771)*** | 1.405 (1.162–1.699)*** | N/A | 1.402 (1.147–1.714)** | N/A | 4.912 (1.130–21.358)* |
| • The sexual behavior styles that I like put me at risk of hepatitis C infection | | N/A | 1.099 (0.970–1.245) | | N/A | | N/A | |
| • I am more worried about hepatitis C than HIV | | N/A | 0.964 (0.845–1.098) | | N/A | | N/A | |
| **Assessment of potential exposures to HCV** | | | | | | | | |
| • Use of intravenous form of recreational drugs | | | | | | | | |
| | No use | 56.9 | Reference | | 60.1 | | 25.7 | |
| | Yes, less than a year | 68.6 | 1.654 (0.799–3.426) | | 82.1 | | 14.3 | |
| | Yes, 1–3 years | 55.0 | 0.927 (0.380–2.262) | | 66.7 | | 20.0 | |
| | Yes, more than 3 years | 15.2 | 0.135 (0.052–0.354) | | 60.0 | | 7.1 | |
| • Engagement in chemosexual behaviors within the preceding 6 months | | | | | | | | |
| | No | 54.4 | Reference | | 60.3 | | 17.0 | |
| | Yes | 64.2 | 1.502 (0.990–2.282) | | 65.7 | | 50.0 | |
| • Status of having a sexual partner within the preceding 6 months | | | | | | | | |
| | No sexual partner | 55.7 | Reference | | 64.3 | | 19.7 | |
| | Regular sexual partner | 53.9 | 0.930 (0.683–1.267) | | 58.1 | | 19.4 | |
| | No regular sexual partners, less than 5 partners | 56.9 | 1.052 (0.707–1.566) | | 59.4 | | 27.3 | |
| | No regular sexual partner, more than 5 partners | 64.0 | 1.414 (0.762–2.625) | | 66.7 | | 0.0 | |
| • Sexual experiences within the preceding 6 months | | | | | | | | |
| | Ever experience sadomasochism | | | | | | | |
| | No | 55.9 | Reference | | 61.3 | | 19.8 | |
| | Yes | 47.8 | 0.723 (0.315–1.657) | | 52.6 | | 25.0 | |
| | Ever experience group sex participation | | | | | | | |
| | No | 55.0 | Reference | | 60.6 | | 20.0 | |
| | Yes | 62.2 | 1.347 (0.843–2.152) | | 64.9 | | 20.0 | |
| | Ever experience Insertive/receptive unprotected anal intercourse | | | | | | | |
| | No | 55.7 | Reference | | 65.0 | | 16.7 | |
| | Yes | 55.7 | 1.000 (0.762–1.312) | | 57.8 | | 28.1 | |
| | Ever experience vaginal sex | | | | | | | |
| | No | 56.3 | Reference | | 61.5 | | 20.6 | |
| | Yes | 50.6 | 0.795 (0.510–1.240) | | 56.8 | | 15.4 | |

Note: *p < 0.05;

**p < 0.01;

***p < 0.001.

Abbreviations: Ab, antibody; Ag, antigen; HBs, hepatitis B surface; HCV, hepatitis C virus; HIV, human immunodeficiency virus; PWID, people who inject drugs; MSM, men who have sex with men; N/A, not available; PLWH, people living with human immunodeficiency virus; UoHCV, unawareness of HCV infection status.

higher education, previously heard of HCV infection, and received information on HCV infection from clinicians (Table 2).

## Development of the HCV knowledge scale among PLWH

The item analysis initially indicated that the original 15 items for measuring HCV knowledge exhibited satisfactory discriminant validity. Exploratory factor analysis finally identified three domains of the 15 items: route of HCV transmission (domain 1), HCV course and complications (domain 2), and HCV treatment (domain 3). Cronbach's α revealed suitable reliability in the three domains, which were thus further analyzed (S1 Table).

## Association between UoHCV and HCV-related knowledge scores stratified by HCV serostatus

Table 3 presents the correct response rates to the 15 HCV-related knowledge items among PLWH with and without awareness of their HCV status, stratified by HCV serostatus. Overall, the correct response rates to the questions ranged from 16.8% to 71.1%. The following three questions had the lowest rates of correct responses: "can hepatitis C virus infection be prevented by vaccines?" (16.8%), "does hepatitis C infection commonly not result in any symptoms?" (31.3%), and "does the successful treatment of hepatitis C virus infection prevent

**Table 3. Comparison of correct responses to structural questions on HCV knowledge between participants with and without awareness of their HCV status, stratified by HCV serostatus.**

| | Correct answer | HCV-seronegative group | | P-value | HCV-seropositive group | | P-value |
|---|---|---|---|---|---|---|---|
| | | Unawareness | Awareness | | Unawareness | Awareness | |
| | | N = 448 | N = 286 | | N = 22 | N = 88 | |
| **Route of HCV transmission** | | | | | | | |
| Does hepatitis C virus can be transmitted through the blood? | Yes | 49.6% | 82.2% | <0.001 | 68.2% | 85.2% | 0.064 |
| Does hepatitis C virus can be transmitted through sexual behaviors? | Yes | 41.7% | 73.8% | <0.001 | 59.1% | 75.0% | 0.138 |
| Does hepatitis C virus can be transmitted through mother-to-child vertical transmission? | Yes | 41.1% | 69.6% | <0.001 | 45.5% | 69.3% | 0.036 |
| Are the infection routes of HIV similar to those of the hepatitis C virus? | Yes | 41.7% | 72.7% | <0.001 | 59.1% | 72.7% | 0.212 |
| During sexual behaviors, does mucosa hemorrhage of sexual contact parts due to excessive intensity makes hepatitis C virus infection easier? | Yes | 40.3% | 76.2% | <0.001 | 57.9% | 68.4% | 0.386 |
| Is blood the major transmission routes of hepatitis C virus? | Yes | 38.8% | 61.5% | <0.001 | 54.5% | 71.6% | 0.125 |
| **Course and complication of HCV** | | | | | | | |
| If you are infected with HIV, does this mean you are more likely to be infected with hepatitis C virus? | Yes | 38.6% | 67.5% | <0.001 | 72.7% | 64.8% | 0.480 |
| Does the successful treatment of hepatitis C virus infection prevent reinfection? | No | 31.5% | 47.2% | <0.001 | 40.9% | 53.4% | 0.294 |
| Hepatitis C virus mostly cures itself, and no treatment is needed | No | 51.6% | 75.2% | <0.001 | 63.6% | 87.5% | 0.008 |
| Does Hepatitis C infection commonly not result in any symptoms? | Yes | 21.4% | 42.3% | <0.001 | 27.3% | 46.6% | 0.101 |
| Do complications after hepatitis C virus infection include cirrhosis and liver cancer? | Yes | 45.5% | 78.0% | <0.001 | 59.1% | 84.1% | 0.01 |
| Does HIV increase complication probability after hepatitis C virus infection (such as cirrhosis and liver cancer)? | Yes | 40.6% | 66.8% | <0.001 | 45.5% | 69.3% | 0.036 |
| **Treatment of HCV** | | | | | | | |
| Can hepatitis C virus infection be prevented by vaccines? | No | 9.2% | 21.7% | <0.001 | 31.8% | 36.4% | 0.690 |
| Can hepatitis C virus infection be treated? | Yes | 57.8% | 83.9% | <0.001 | 68.2% | 97.7% | <0.001 |
| Can Hepatitis C virus infection be cured? | Yes | 33.9% | 53.5% | <0.001 | 40.9% | 83.0% | <0.001 |

Abbreviations: HCV, hepatitis C virus; HIV, human immunodeficiency virus.

reinfection?" (39.3%). The question "can hepatitis C virus infection be treated?" had the highest proportion of correct responses (71.1%).

Multilinear regression analyses revealed that UoHCV was associated with lower mean scores overall and for each domain of HCV-related knowledge compared with HCV status awareness, both in the HCV-seronegative group and the HCV-seropositive group (Table 4).

## Discussion

To the best of our knowledge, ours is the first study to reveal differences in the prevalence rates of UoHCV among PLWH; these rates were 61.0% in the HCV-seronegative group and 20.0% in the HCV-seropositive group. The prevalence rate in the HCV-seropositive group increased to 33.8% after the exclusion of participants with a history of HCV treatment, all of whom knew their HCV status. After stratification by HCV serostatus, the two groups differed in terms of their sociodemographic characteristics and laboratory variables associated with UoHCV, indicating the need for strategies to be tailored according to HCV serostatus when attempting to reduce UoHCV among PLWH.

Our results revealed a higher prevalence of UoHCV among MSM than among PWID (25.9% vs.7.3%, p = 0.037) in the HCV seropositive population. Since the 1990s, several measures have been implemented to promote HCV testing and to enhance the awareness of HCV infection status among PWID. These measures have achieved significant effects in increasing HCV status awareness [18, 38, 39]. The risk of sexual transmission of HCV was low in the 1990s [40]. However, since the mid-2000s, sexually transmitted HCV has been increasingly detected among sexually active MSM [6, 20]. The high prevalence of UoHCV among MSM may be attributable to their unawareness of their HCV seropositivity status due to the fundamental misunderstanding that HCV can only be contracted from intravenous exposure to infected blood (e.g., through the use of unsterile injection equipment and contaminated blood products). This explanation is supported by the finding that individuals in this study who believed that intravenous injection was a requirement for contracting HCV were more likely to have UoHCV.

**Table 4. Association of UoHCV with HCV knowledge scores, stratified by knowledge domain in a multivariable linear regression[†].**

| | *Total score* | *Domain 1 (Route of HCV transmission)* | *Domain 2 (HCV course and complications)* | *Domain 3 (HCV treatment)* |
|---|---|---|---|---|
| | *Adjusted Model Estimate (β) (95% CI)* | *Adjusted Model Estimate (β) (95% CI)* | *Adjusted Model Estimate (β)(95% CI)* | *Adjusted Model Estimate (β) (95% CI)* |
| Total participants | | | | |
| • Unawareness of HCV infection status | -0.232 (-0.271–-0.192)*** | -0.271 (-0.321–-0.220)*** | -0.217 (-0.262–-0.173)*** | -0.190 (-0.232–-0.147)*** |
| HCV-seronegative group | | | | |
| • Unawareness of HCV infection status | -0.239 (-0.281–-0.197)*** | -0.283 (-0.337–-0.228)*** | -0.222 (-0.270–-0.175)*** | -0.182 (-0.227–-0.137)*** |
| HCV-seropositive group | | | | |
| • Unawareness of HCV infection status | -0.195 (-0.305–-0.084)** | -0.189 (-0.337–-0.040)* | -0.177 (-0.301–-0.054)** | -0.282 (-0.404–-0.160)*** |

Note: *p < 0.05;

**p < 0.01;

***p < 0.001.

[†]Adjustments were made in the multilinear regression for the period since participants received their HIV diagnosis and for their gender, age, education level, employment status, marital status, site of HIV diagnosis, HIV risk factors, history of sexually transmitted diseases in the preceding 6 months, and history of HCV therapy.

Abbreviations: Ab, antibody; CI, confidence interval; HCV, hepatitis C virus.

Furthermore, an increasing trend of MSM was observed in the HCV-seropositive PLWH group (28.3%, 68.0%, and 81.3% in periods 1, 2, and 3, respectively [p for trend < 0.001]; S2 Table), which is consistent with findings of another study conducted in Taiwan [41]. Therefore, substantial efforts are urgently required to implement interventions that reduce the prevalence rate of UoHCV among HIV-positive individuals with sexually transmitted HCV.

The lower prevalence of UoHCV among HCV-seropositive PLWH diagnosed in 2014–2020 compared with the prevalence among those diagnosed before 2008 may be attributable to the results of recent efforts to fight the spread of HCV. These efforts include the increase in publicity on HCV worldwide, newly available DAAs, and implementation of measures to ensure the affordability of oral DAA agents in Taiwan; such measures include national programs for using DAA agents to treat patients with advanced HCV (implemented in 2017–2018) and for providing treatment support to all eligible patients enrolled in Taiwan's National Health Insurance program (2019–present) [42, 43]. This means that compared with before 2008, there is now a higher likelihood of primary physicians testing for HCV and identifying HCV seropositivity in people recently diagnosed as having HIV and informing them of their condition. Moreover, primary physicians may be more likely to inform individuals of their HCV-seropositive status when diagnosing them as having a coinfection with sexually transmitted HAV because of the emerging trend of concomitant sexually transmitted HAV and HCV among PLWH in Taiwan [41, 44].

Reducing the engagement in high-risk behavior is critical for reducing rates of HCV infection. People who have received treatment for HCV who have ongoing exposure to HCV are at risk of reinfection. Limiting such exposure is necessary for making continual progress toward eradication of the disease [45]. Studies have revealed that awareness of HCV infection status reduces engagement in high-risk behaviors among HCV-seropositive PWID [9–12]. However, a significant association was not observed between UoHCV and high-risk behaviors related to HCV infection in the present study, in which the majority (94.8%) of participants had sexually transmitted HIV infection. This finding has two possible explanations. First, individuals who were aware of their HCV status may not have known its transmission route (e.g., sexual transmission). Second, individuals may have known the transmission route but been unwilling or unable to alter their behavior accordingly. Our findings support the second explanation because the participants with awareness of their HCV-seropositive status had higher scores in knowledge domain 1 (route of HCV transmission) than those without awareness of their HCV-seropositive status, indicating that they did not alter their behavior despite knowing the HCV transmission route. The results of a cross-sectional online survey of 48 Australian MSM further support our findings. The survey revealed that most participants knew that HCV infection can be sexually transmitted between men. However, participants generally did not know how to prevent the sexual transmission of HCV [46]. Although notifying PWID of their HCV status can effectively reduce their engagement in high-risk behaviors [9–12], effective behavior modification approaches for reducing the sexual transmission of HCV in at-risk populations have yet to be identified. Moreover, interventions to reduce modifiable behavioral risk factors, such as condom distribution, the promotion of abstinence from illicit drugs, and advocacy for safe sex, may attract serious criticism because of the stigma associated with sex and drug use [47]. Therefore, to optimize the awareness of the HCV care cascade among HCV-seropositive PLWH [8], further research is necessary to identify effective interventions for modifiable behavioral risk factors and settings during the process of notifying patients of their HCV status to promote long-term reductions in behaviors that involve a high-risk of exposure to HCV [48].

In the HCV-seronegative population, the prevalence of UoHCV (61.0%) among PLWH reported in the present study is substantially higher than the prevalence of general population

reported by a nationwide screening program in Taiwan (33.0%) [16], possibly because of the high proportion of MSM (76.6%) who believe that intravenous injection is a requirement for contracting HCV.

In the present study, only 288 (39.2%) of the 734 HCV-seronegative PLWH responded "yes" to the question "has your doctor ever provided information regarding HCV infection?" This number was unexpectedly low considering that these PLWH were undergoing follow-up every 1–3 months at HIV referral hospitals in Taiwan. According to Taiwan's guidelines for HIV and acquired immunodeficiency syndrome, HCV-seronegative PLWH should undergo annual screening for HCV antibodies [49]. HIV experts may overlook the importance of informing PLWH of their HCV-seronegative status. However, our data revealed that UoHCV is associated with lower mean scores for total and domain-specific HCV-related knowledge compared with HCV status awareness among PLWH with HCV-seronegative status. Furthermore, young, sexually active PLWH may primarily consult and receive serostatus notifications of HCV infection from HIV experts because of the stigma and discrimination that PLWH may experience when consulting with non-HIV experts [50]. Therefore, HIV experts should notify sexually active PLWH of their HCV-seronegative status. This is especially crucial for individuals who believe that HCV can only be transmitted between PWID or who have been diagnosed as having a sexually transmitted disease within the preceding 6 months.

Overall, our study revealed that the correct response rate for questions on HCV-related knowledge varied considerably (16.8% to 71.1%). This finding provides a basis for targeting gaps in patient HCV knowledge during counseling. In our study, UoHCV was determined to be negatively associated with different aspects of HCV-related knowledge, which is considered critical for initiating treatment for HCV infection [28, 51]. The findings of the present study also indicate that interventions tailored to each patient's HCV serostatus should be actively enforced to reduce the prevalence of UoHCV.

The present study has several strengths. This is the first study to analyze UoHCV among PLWH and identify different determinants associated with UoHCV among PLWH according to their HCV serostatus. The study results also aid in the customization of strategies according to their HCV serostatus for reducing the prevalence of UoHCV among PLWH. Second, no standardized, validated HCV knowledge scoring system for PLWH is currently available. This study provides clinically relevant, structural, and valid measurements of HCV-related knowledge among PLWH. This structural measure of HCV-related knowledge can be applied to assess HCV-related knowledge before and after HCV-related educational interventions. These findings can also be used to longitudinally assess the influence of knowledge regarding HCV transmission on patients' engagement in high-risk behaviors associated with HCV infection after they have completed treatment interventions. However, this study also has several limitations. First, 525 patients were unwilling to participate, which might have caused selection bias. However, the demographic characteristics, HCV serostatus, and HIV-related risks did not differ between the enrolled and unwilling patients. Second, although the trained investigators instructed participants to recall their experiences to answer certain questions, inaccurate recall was unavoidable. Third, although the current study revealed no association between high-risk behaviors for HCV infection and UoHCV among HCV-seropositive and HCV-seronegative PLWH, this study had a cross-sectional design. Therefore, additional prospective cohort studies are warranted to further clarify changes in individuals' engagement in high-risk behaviors after they have been notified of their HCV status. Finally, the participants in the present study were PLWH, most of whom were MSMs. Therefore, the findings may not be generalizable to all PLWH populations.

## Conclusion

With the current availability of all-oral, interferon-free DAAs, which achieve a curative out-come in more than 90% of patients with HCV and require only 12 weeks of treatment, the eradication of HCV in prevalent areas is now possible [52]; however, our study revealed that the majority of PLWH in Taiwan were unaware of their HCV serostatus. In the present study, UoHCV was associated with lower scores in different domains of HCV-related knowledge, which is considered critical for initiating treatment for HCV infection and reducing the engagement in risky behaviors among affected individuals. This finding suggests that interventions should be implemented to reduce the prevalence of UoHCV, regardless of patients' HCV serostatus. Because the factors associated with UoHCV differed between HCV-seronegative and HCV-seropositive PLWH, strategies for reducing UoHCV may be more effective if they target PLWH according to their HCV serostastus.

## Supporting information

**S1 File. Questionnaire on HCV infection status and related knowledge among patients living with HIV.**
(DOCX)

**S1 Table. Validation of the original 15 questions on HCV knowledge.**
(DOCX)

**S2 Table. Trends of various HIV at-risk populations among HCV-seropositive patients living with HIV across three periods of HIV diagnosis.**
(DOCX)

## Acknowledgments

The authors would like to thank Wallace Academic Editing for the English language review.

## Author Contributions

**Conceptualization:** Chun-Yuan Lee, Tun-Chieh Chen.

**Data curation:** Chun-Yuan Lee, Pei-Hua Wu, Meng-Wei Lu, Tun-Chieh Chen.

**Formal analysis:** Chun-Yuan Lee, Pei-Hua Wu, Meng-Wei Lu.

**Funding acquisition:** Chun-Yuan Lee.

**Investigation:** Chun-Yuan Lee, Tun-Chieh Chen.

**Methodology:** Chun-Yuan Lee, Pei-Hua Wu, Meng-Wei Lu, Tun-Chieh Chen, Po-Liang Lu.

**Supervision:** Po-Liang Lu.

**Validation:** Po-Liang Lu.

**Writing – original draft:** Chun-Yuan Lee, Pei-Hua Wu.

**Writing – review & editing:** Pei-Hua Wu, Meng-Wei Lu, Po-Liang Lu.

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
