## [Decision Letter · Decision Letter 0]

18 Mar 2021

PONE-D-20-37478

High prevalence of unawareness of HCV infection status among both anti-HCV seronegative and seropositive people living with human immunodeficiency virus in Taiwan

PLOS ONE

Dear Dr. Lu,

Thank you for submitting your manuscript to PLOS ONE. After careful consideration, we feel that it has merit but does not fully meet PLOS ONE’s publication criteria as it currently stands. Therefore, we invite you to submit a revised version of the manuscript that addresses the points raised during the review process.

ACADEMIC EDITOR: Please note that the Academic Editor is Reviewer #2, and provided significant input to help clarify issues in the manuscript. The paper has many strengths and addressing these will significantly strengthen th paper.  Please have the English reviewed again before resubmission.

We look forward to receiving your revised manuscript.

Kind regards,

Kimberly Page, PhD, MPH

Academic Editor

PLOS ONE

Journal Requirements:

2, We note that you have included the phrase “data not shown” in your manuscript. Unfortunately, this does not meet our data sharing requirements. PLOS does not permit references to inaccessible data. We require that authors provide all relevant data within the paper, Supporting Information files, or in an acceptable, public repository. Please add a citation to support this phrase or upload the data that corresponds with these findings to a stable repository (such as Figshare or Dryad) and provide and URLs, DOIs, or accession numbers that may be used to access these data. Or, if the data are not a core part of the research being presented in your study, we ask that you remove the phrase that refers to these data.

Additional Editor Comments (if provided):

Thank your for submitting your manuscript. It addressed an very important issue - the first step in the HCV cascade of care. Please note that the I as Academic Editor conducted the second review.

Reviewers' comments:

Reviewer's Responses to Questions

**Comments to the Author**

1. Is the manuscript technically sound, and do the data support the conclusions?

Reviewer #1: Yes

Reviewer #2: Yes

2. Has the statistical analysis been performed appropriately and rigorously? 

Reviewer #1: Yes

Reviewer #2: Yes

3. Have the authors made all data underlying the findings in their manuscript fully available?

Reviewer #1: Yes

Reviewer #2: Yes

4. Is the manuscript presented in an intelligible fashion and written in standard English?

Reviewer #1: Yes

Reviewer #2: Yes

5. Review Comments to the Author

Reviewer #1: Hepatitis C elimination is a global public health task. This research has addressed one important issue in this challenging program. Some comments/recommendations for the authors:

1. When describe the time period, “<2008” might be better changed into “before 2008”.

1.1 In the manuscript, “The participants were categorized into three distinct periods based on the date of 127 HIV diagnosis: before 2008 (period 1), 2008–2013 (period 2), and 2014–2020 (period 3).” – it meant the first confirmed diagnosis of HIV? The earliest date of diagnosis in the study hospitals? Patients do not overlap in the 3 periods. Please clearly describe if possible.

1.2 It seems the authors separate the 3 groups of HIV patients to imply different cohorts might have different (unawareness) of HCV. However, the title of Figure 2 may have misleading wordings. Recommend using more precise wordings.

2 Page 13, Column 3, Line 5 – “360 ()80.4” should be corrected.

3 Page 13, Column 3, Line 6 – “21 (4,7)” should be corrected.

4 Page 15, First line – “Unawareness of HCV infection status…” is not applicable (necessary) for analysis.

5 Page 39, First line – the answer for that is “Yes” not “Blood”.

Reviewer #2: Editor's Comments

This is an interesting paper that has some strengths, including the overall study question, the target population and the setting. Taiwan has a significant burden of HCV. With current DAA treatments which offer cure over 90% of those treated, awareness of infection is the first step to getting people engaged in that beneficial care. There are numerous weaknesses and editorial issues that the authors need to address before this is suitable for publication. There do remain many areas for English grammar and syntax improvement as well. Below I list these, including both major and minor issues, by paper section.

ABSTRACT

1. Line 11-12. Please clarify if unawareness was that the participant incorrectly identified being anti-HCV negative if they were truly anti-HCV positive and if they identified as anti-HCV positive but were truly anti-HCV negative. This will need to be made clear later in the manuscript as well.

2. Line 17: I think it would be more appropriate to use the term Men who have sex with men (MSM) rather than homosexual. This should also be changed throughout other parts of the manuscript as well.

3. Line 17: Pleas change the term 'injection drug users" to "people who inject drugs" (PWID) here and throughout the manuscript.

4. Line 10: less hearing of HCV before. I think what is meant is that few had previously heard of HCV infection prior to the research. If this is correct, please change it. (This is an English issue)

INTRODUCTION

1. Lines29-30: The term "Elimination" should probably be replaced. HCV Elimination is used as a term in public health referring to population level control.No one has achieved elimination from a public health perspective. I think that authors mean to say that treatment with DAAs results in cure in 90% or more of patients (i.e. it is true that virus is eliminated in the patients body). I suggest reframing this about cure (more patient centered).

If the authors want - they can note that treatment with DAAs 'can lead to elimination of HCV viremia and cure.

2. Line 33: this statement should include a reference/citation regarding reinfections among people who use drugs, (were highest incidence is)

3. line 35: change "among' to "to"

4. Line 43: regarding "41% in Germany" - this reference is very dated and this may have changed since 2014 when DAAs became available. Please update.

5. Line 46: Please change intravenous drug users (IDUs) to People who inject drugs (PWID) here and throughout the manuscript.

6. Line 45-46 - regarding change in behavior with awareness of HCV status - not all research shows that this happens. Consider these: https://pubmed.ncbi.nlm.nih.gov/25814695/
https://www.sciencedirect.com/science/article/abs/pii/S0376871609002294?via%3Dihub

7. Line 51: The only references cited for this statement are with respect to sexual transmission. Please add references regarding parenteral transmission, which is FAR more common than sexual transmission.

8. Line 52: This reference is with respect to India, what what you cite is true. However HIV is highly stigmatized there. Strengthen this statement with some data from a different area/region.

9. Line 53: add: "especially marginalized groups, which may contribute to a significantly decreased..."

10. Lines 54-55 - i think caution should be used when prioritizing one population over another. This has the effect of putting value on one population over another. It may be enough to state that this is a high priority population and cite guidelines for treatment of HCV in PLWH from Infectious Disease Society, and or other clinical groups.

You can consider deleting the word 'more' and 'than in the general population' in this line.

11. Line 58-59 I think this sentence is too strong and a little off the mark: awareness of serostatus would facilitate the next step in continuum of care - linking to a provider and potentially treatment, which are all necessary steps to elimination.

12. Line 70. Please describe the Domains.

MATERIALS AND METHODS

1. Line 80: in the last sentence that says Southern Taiwan has a high prevalence of HCV infection, please add: 'relative to -----where and how much--. This will give the reader more perspective.

2. Linnes 84-85: please clarify what the difference is between "perceived risk of HCV infection, and assessment of the risk of HCV infections". (after reading the survey and the results the latter is about 'potential exposures to HCV". I do recommend using this language about exposures vs. risk elsewhere in the manuscript.

3. Lines 96-97 - Again, I think the authors should consider changing the language for the last group to "assessment of potential exposures to HCV".

4. Lines 127-128: please state why these time periods were selected.

5. Lines 120-131: This English is a little awkward. Consider changing to: 'correctly identifying as anti-HCV positive', and UoHCV as 'incorrectly identifying as anti-HCV negative"

6. Line 131: change high risk behaviors to high risk exposures (it is less judgmental).

7. Line 134: what does 'status' refer to? the HCV status of sexual partners? Please clarify.

8. Line 142: again, what are the Domains - please describe these.

9. Line 152: Please omit the word univariate and use the word bivariate when describing analyses comparing two variables. (univariate describes one variable).

10: Line 153Clarify what statistic was used to assess bivariate associations (it is not clear why use binary backwards logistic regression?? ). this sentence should probably read: Binary logistic Binary regression was used to assess associations and calculate odds ratios in bivariate analyses between surveyed factors and UoHCV... in .

11. Line 163: Consider editing this to: To determine HCV-related knowledge and domain scores of knowledge independently associated with UoHCV, we used a multilinear regressions model with backward strategy. Also, please clarify why Backward approach was used.

RESULTS

1. Lines 176-177: State what the subgroups are in the text.

2. Table 2: please change the word 'univariable' to Bivariate in the title and in the Table.

3. Lines 245-247 - the domains noted here should be moved to methods. It is find to reiterate them here as well.

4. Table 4. Please Name the Domains, not just the number. in the table heading. Can you convert the Betas and 95% CI to adjusted odds ratios.

DISCUSSION

1. Line 276: Be very sure of this... it may be the first in a study of HIV positive people (but its not the first study to assess discordance in HCV status and knowledge of status.)

2. Lines 282-283: what does this mean (need for customized strategies). This sentence is too broad - customized to who? or should messages be broadened to other groups. Be more specific .

3. Line 285 and elsewhere: change IDU to PWID.

4. Line 290: delete 'prevalent among" and change to "increasingly detected"

5. Line 291 and elsewhere: change homosexuals to MSM.

6. Line 294: delete medical, and replace with injection

7. Line 295: delete "or sharing of unsterile equipment" as it is redundant with the above.

8. Line 302: add "HIV positive" before the word 'individuals'

9. Line 306: Add "newly available DAAs" to this sentence.

10. Lines 316-318: Reframe this sentence. There is risk of reinfection among people treated with HCV who have ongoing exposures. Limiting these is needed to maintain elimination progress.

11. Line 320 - others researchers have not seen behavior change: (i) https://www.sciencedirect.com/science/article/abs/pii/S0376871609002294?via%3Dihub

(ii) https://jech.bmj.com/content/69/8/745

But note that these studies were not conducted in PLWH

12. Line 336: When you say 'further research' be more engaged. Like what research is needed.

13. Lines 337-339: This is a very old reference. Many attitudes have changed. Does stigma remain high in Taiwan? Perhaps local reference might be better here to better frame what is needed in Taiwan.

14. Lines 341-342: delete part of the sentence that says: ..."particularly among individuals... before 2008"

15. Line 343: When saying additional study, should be conducted - like what ? besides condom use?

16. Line 354-355". What is the medical guideline for HCV testing in HIV positive people in Taiwan? this would be good information to share and make the point that the HCV testing rate is low in this high risk group.

17. Lines 357-358" The sentence states there is a negative association: please interpret this for our readers. What does this mean?

18. Several places -where Tables are referred to in the Discussion. I think you do not have to tell the reader to go back to the tables and can delete those.

19. Line 376: please make sure to state the population a the end of the sentence ".. in PLWH".

20: Line 388: Good job on the limitations. I suggest deleting the words "was unavoidable" from this sentence though.

21. Line 394 change this sentence to read: ".. findings may not be generalizable to all PLWH populations. [i feel that in many places the authors lose sight that this is all done in PLWH - so discussion points need to emphasize that].

CONCLUSION

1. Line 399" say why the 'era of DAA's is important (eg., the high cure rates and potential to impact population prevalence or lead to elimination...?)

2. Line 405: Delete the end of the sentence starting with 'should' and change to "may be more effective if targeted to PLWH by HCV serostastus. (stay with your data and your population'. (the word personalized is awkward)

6. PLOS authors have the option to publish the peer review history of their article (what does this mean?). If published, this will include your full peer review and any attached files.

Reviewer #1: No

Reviewer #2: No

---

## [Author Response · Author response to Decision Letter 0]

11 Apr 2021

Journal Requirements

Response:

We have revised it as PLOS ONE's style requirements.

Response:

We have added a new supplementary table 2 (S2_Table) in the revised manuscript. 

Responses to the comments of the Reviewer 1

1. When describe the time period, “<2008” might be better changed into “before 2008”.

Answer:

Thank for the reviewer’s suggestion. We have corrected it throughout the manuscript.

2. In the manuscript, “The participants were categorized into three distinct periods based on the date of 127 HIV diagnosis: before 2008 (period 1), 2008–2013 (period 2), and 2014–2020 (period 3).” – it meant the first confirmed diagnosis of HIV? The earliest date of diagnosis in the study hospitals? Patients do not overlap in the 3 periods. Please clearly describe if possible.

Answer:

Thank for the reviewer’s suggestion, and we have revised it as “The participants were categorized by three distinct periods based on the calendar year of their first confirmed HIV diagnosis: before 2008 (period 1, before the remission of the HIV epidemic among PWID) [36], 2008–2013 (period 2, remission of the HIV epidemic among PWID but before the introduction of oral DAAs), and 2014–2020 (period 3, after the introduction of oral DAAs).” Materials and Methods section, line 139-143.

3. It seems the authors separate the 3 groups of HIV patients to imply different cohorts might have different (unawareness) of HCV. However, the title of Figure 2 may have misleading wordings. Recommend using more precise wordings.

Answer:

We have revised the title of figure 2 as “Fig 2. Trend analyses of prevalence of UoHCV stratified by HCV serostatus in three periods according to the calendar year of first confirmed HIV diagnosis ( period 1 [before 2008], period 2 [2008–2013], and period 3 [2014–2020]).” Result section, line 235-237.

4. Page 13, Column 3, Line 5 – “360 ()80.4” should be corrected.

Answer:

We have corrected it.

5. Page 13, Column 3, Line 6 – “21 (4,7)” should be corrected.

Answer:

We have corrected it.

6. Page 15, First line – “Unawareness of HCV infection status…” is not applicable (necessary) for analysis

Answer:

Thank for the reviewer’s suggestion, and we have deleted it.

7. Page 39, First line – the answer for that is “Yes” not “Blood”.

Answer:

Thank for the reviewer’s suggestion, and we have corrected it. 

Responses to the comments of the Reviewer 2 (Academic editor)

ABSTRACT

1. Line 11-12. Please clarify if unawareness was that the participant incorrectly identified being anti-HCV negative if they were truly anti-HCV positive and if they identified as anti-HCV positive but were truly anti-HCV negative. This will need to be made clear later in the manuscript as well.

Answer:

Thank for the editor’s valuable suggestion. We will explain below:

In the manuscript under the Method section, we mentioned “awareness of HCV infection status was defined as participants’ self-reported recognition of their HCV infection status at the time of enrollment in the study (i.e., HCV-seropositive patients’ awareness of their HCV-positive status and HCV-seronegative patients’ awareness of their HCV-negative status), whereas UoHCV was defined as participants’ self-reported unawareness of their HCV infection status”. Materials and Methods section, line 144-149.

Therefore, for HCV seropositive group, awareness of HCV infection status mean that the individual recognize himself/herself as HCV-positive status (whether the awareness is correct or incorrect); for HCV seronegative group, awareness of HCV infection status mean that the individual recognize himself/herself as HCV-negative status.

However, the awareness of self-HCV infection status may be correct or incorrect. According to the study conducted by Cheryl et al. (J Glob Health . 2019 Jun;9(1):010426), the rate pf correctly aware of their HCV infection status was 44.6% for anti-HCV (+) and 66.3% for anti-HCV (-). However, the issue of the incorrect/correct awareness is not the goal of the present study. We appreciate the editor’s comments, and we may discuss it in the next study.

To clarify it in the abstract, we revised it as “HCV infection status awareness is crucial in the HCV care continuum for both HCV-seropositive (HCV-positive status awareness) and seronegative (HCV-negative status awareness) populations. However, trends in the unawareness of HCV infection status (UoHCV) remain unknown in HIV-positive patients. This study investigated UoHCV prevalence, the associated factors of UoHCV, and its association with HCV-related knowledge in HIV-positive patients.” Abstract section, line 2-7.

2. Line 17: I think it would be more appropriate to use the term Men who have sex with men (MSM) rather than homosexual. This should also be changed throughout other parts of the manuscript as well.

Answer:

Thank for the editor’s suggestion, and we have corrected it throughout the manuscript.

3. Line 17: Pleas change the term 'injection drug users" to "people who inject drugs" (PWID) here and throughout the manuscript.

Answer:

Thank for the editor’s suggestion, and we have corrected it throughout the manuscript.

4. Line 10: less hearing of HCV before. I think what is meant is that few had previously heard of HCV infection prior to the research. If this is correct, please change it. (This is an English issue).

Answer:

Thank for the editor’s suggestion, and we have corrected it as “participants with UoHCV were more likely to have a recent history of sexually transmitted diseases, but had a lower education level, had received less information on HCV infection from clinicians, and were less likely to have heard of HCV infection prior to the research”. Abstract section, line 21-24.

INTRODUCTION

1. Lines29-30: The term "Elimination" should probably be replaced. HCV Elimination is used as a term in public health referring to population level control. No one has achieved elimination from a public health perspective. I think that authors mean to say that treatment with DAAs results in cure in 90% or more of patients (i.e. it is true that virus is eliminated in the patients body). I suggest reframing this about cure (more patient centered).

If the authors want - they can note that treatment with DAAs 'can lead to elimination of HCV viremia and cure.

Answer:

Thank for the editor’s valuable suggestion. We have reframed it as “Although treatment with direct-acting antivirals (DAAs) can lead to the elimination of HCV viremia and a curative outcome in more than 90% of patients with chronic HCV infection,…..”. Introduction section, line 32-34.

2. Line 33: this statement should include a reference/citation regarding reinfections among people who use drugs, (were highest incidence is)

Answer:

Thank for the editor’s valuable suggestion. We have revised it as “several barriers to eradicating HCV infection still exist, including the high costs of drugs [3], frequent loss to follow-up after diagnosis [4], high rate of early HCV reinfection among patients who have recently received drug injections [5], and ongoing high-risk behaviors associated with HCV infection (even after clearance of HCV infection) [6].” Introduction section, line 34-38.

3. line 35: change "among' to "to"

Answer:

We have corrected it.

4. Line 43: regarding "41% in Germany" - this reference is very dated and this may have changed since 2014 when DAAs became available. Please update.

Answer:

Thank for the editor’s suggestion. However, we didn’t find out another updated reference about the prevalence of awarensss of self-HCV infection status in German in the era of DAAs. Therefore, we deleted this reference.

5. Line 46: Please change intravenous drug users (IDUs) to People who inject drugs (PWID) here and throughout the manuscript.

Answer:

Thank for the editor’s suggestion, and we have corrected it throughout the manuscript.

6. Line 45-46 - regarding change in behavior with awareness of HCV status - not all research shows that this happens. Consider these: https://pubmed.ncbi.nlm.nih.gov/25814695/
https://www.sciencedirect.com/science/article/abs/pii/S0376871609002294?via%3Dihub

Answer:

We thank for the editor’s valuable suggestions. The two studies aimed to evaluate the behavior change after notification of HCV infection status. 

Brief introduction of the two studies were listed below:

1. https://www.sciencedirect.com/science/article/abs/pii/S0376871609002294?via%3Dihub:

This prospective study of 112 young PWID who experienced HCV seroconversion during follow-up demonstrated a modest reduction on alcohol and non-injection drug use that is not sustained over time after the young IDU were aware of their HCV seroconversion.

2. https://pubmed.ncbi.nlm.nih.gov/25814695/

This within-cohort matching study enrolled 190 HCV-positive and HCV-negative participnts at baseline. This study demonstrate a 5% per 3-month reduction post-notification in the odds of recent injection drug use (adjusted Odds Ratio: aOR 0.95, 95% CI 0.93–0.96) in HCV-positive group and a 8% per 3-month reduction post-notification in the odds of recent syringe borrowing (adjusted Odds Ratio: aOR 0.92, 95% CI 0.87–0.97) in HCV-negative group.

Therefore, we revised the paragraph as “Although the short- and long-term impacts of HCV-positive status awareness among HCV-seropositive patients on their risk behavior remain matters of debate [9-12], HCV-positive status awareness is essential in the HCV care continuum in terms of treatment eligibility and taking medical advice on viral transmission [12,13].” Introduction section, line 43-47. and as “In the HCV-seronegative population at risk of contracting HCV, people who inject drugs (PWID) may engage in high-risk behaviors (e.g., sharing a syringe or injecting themselves with drugs) less frequently if they are aware of their HCV infection status (i.e., HCV-negative status awareness) [9,12].” Introduction section, line 50-53.

7. Line 51: The only references cited for this statement are with respect to sexual transmission. Please add references regarding parenteral transmission, which is FAR more common than sexual transmission.

Answer:

Thank for the editor’s suggestion. We have added an another reference 21 (PLoS One. 2019 Dec 10;14(12):e0226166), as below:

“People living with human immunodeficiency virus (HIV) infection (PLWH) are at risk of HCV infection because the transmission routes of HCV infection, such as unprotected sex and drug injection, are similar to those of HIV infection [6,20,21].” Introduction section, line 56-58.

8. Line 52: This reference is with respect to India, what you cite is true. However HIV is highly stigmatized there. Strengthen this statement with some data from a different area/region.

Answer:

Thank for the editor’s suggestion, and we have added another two reference 23 (Drug Alcohol Depend. 2008 Jan 11;93(1-2):141-7) and 24 (J Acquir Immune Defic Syndr

. 2004 Nov 1;37(3):1367-75), and revised it as “Moreover, individuals coinfected with HCV and HIV are less likely to seek HCV care [22-24].” Introduction section, line 59-60.

9. Line 53: add: "especially marginalized groups, which may contribute to a significantly decreased..."

Answer:

Thank for the editor’s suggestion, and we have revised it as “Moreover, individuals coinfected with HCV and HIV are less likely to seek HCV care [22-24], which may contribute to a significantly decreased quality of life and quicker progression of liver disease, especially in those who are homeless or marginally housed [25].” Introduction section, line 59-62.

10. Lines 54-55 - I think caution should be used when prioritizing one population over another. This has the effect of putting value on one population over another. It may be enough to state that this is a high priority population and cite guidelines for treatment of HCV in PLWH from Infectious Disease Society, and or other clinical groups.

You can consider deleting the word 'more' and 'than in the general population' in this line.

Answer:

Thank for the editor’s valuable suggestion, and we have revised it as “Additionally, patients with HCV/HIV coinfection have higher rates of death and disease progression, including the progression of histological fibrosis/cirrhosis and decompensated liver disease, than do patients with HCV monoinfection [26]. Therefore, HCV screening, treatment, and prevention strategies should be strictly implemented among PLWH [27].” We also added a new reference 27: Guidelines for Adults and Adolescents. Guidelines for the Use of Antiretroviral Agents in Adults and Adolescents Living with HIV. 2021. Introduction section, line 62-66.

11. Line 58-59 I think this sentence is too strong and a little off the mark: awareness of serostatus would facilitate the next step in continuum of care - linking to a provider and potentially treatment, which are all necessary steps to elimination.

Answer:

Thank for the editor’s suggestion, and we have revised it as “Serostatus awareness facilitates the next step in the continuum of HCV care, namely providing affected patients with access to health care, relevant consultation, and potential treatment, which are necessary to eradicate HCV.” Introduction section, line 68-71.

12. Line 70. Please describe the Domains.

Answer:

We have revised it as “We evaluated the prevalence of UoHCV, explored the determinants of UoHCV, and evaluated the associations of UoHCV with different domains of HCV-related knowledge (i.e., route of HCV transmission, HCV course and complications, and HCV treatment) among a sample of PLWH stratified by HCV serostatus.” Introduction section, line 80-84.

MATERIALS AND METHODS

1. Line 80: in the last sentence that says Southern Taiwan has a high prevalence of HCV infection, please add: 'relative to -----where and how much--. This will give the reader more perspective.

Answer:

We have revised it as “The HCV seropositivity in southern Taiwan is 8.6% [31], which is higher than that in northern Taiwan (1.2%–2.7%) [32].” Materials and Methods section, line 92-93.

2. Lines 84-85: please clarify what the difference is between "perceived risk of HCV infection, and assessment of the risk of HCV infections". (after reading the survey and the results the latter is about 'potential exposures to HCV". I do recommend using this language about exposures vs. risk elsewhere in the manuscript.

Answer:

Thank for the editor’s suggestion, and we have revised it throughout the manuscript.

3. Lines 96-97 - Again, I think the authors should consider changing the language for the last group to "assessment of potential exposures to HCV".

Answer:

Thank for the editor’s suggestion. We have revised it throughout the manuscript and the S1 file.

4. Lines 127-128: please state why these time periods were selected.

Answer:

We have revised it as “The participants were categorized by three distinct periods based on the calendar year of their first confirmed HIV diagnosis: before 2008 (period 1, before the remission of the HIV epidemic among PWID) [36], 2008–2013 (period 2, remission of the HIV epidemic among PWID but before the introduction of oral DAAs), and 2014–2020 (period 3, after the introduction of oral DAAs).” Materials and Methods section, line 139-143.

5. Lines 120-131: This English is a little awkward. Consider changing to: 'correctly identifying as anti-HCV positive', and UoHCV as 'incorrectly identifying as anti-HCV negative"

Answer:

As we have mentioned earlier. In the manuscript under the Method section, we defined awareness of HCV infection status as “participants’ self-reported recognition of their HCV infection status at the time of enrollment in the study (i.e., HCV-seropositive patients’ awareness of their HCV-positive status and HCV-seronegative patients’ awareness of their HCV-negative status). Materials and Methods section, line 144-147.

Therefore, for HCV seropositive group, awareness of HCV infection status mean that the individual recognize himself/herself as HCV-positive status (whether the awareness is correct or incorrect); for HCV seronegative group, awareness of HCV infection status mean that the individual recognize himself/herself as HCV-negative status.

To clarify it, we have revised it as “In this study, awareness of HCV infection status was defined as participants’ self-reported recognition of their HCV infection status at the time of enrollment in the study (i.e., HCV-seropositive patients’ awareness of their HCV-positive status and HCV-seronegative patients’ awareness of their HCV-negative status), whereas UoHCV was defined as participants’ self-reported unawareness of their HCV infection status [8,16].” Materials and Methods section, line 144-149.

6. Line 131: change high risk behaviors to high risk exposures (it is less judgmental).

Answer:

Thank for the editor’s suggestion, and we have revised it as “The behavioral indicators of a high risk of exposure to HCV infection were modified from other studies and included using any intravenous recreational drugs [12], engaging in chemosexual behaviors within the preceding 6 months [37], having a sexual partner within the preceding 6 months (assessment options were no sexual partners, one regular sexual partner, no regular sexual partners/less than five partners, and no regular sexual partner/more than five partners) [37], and engaging in other activities involving sexual contact within the preceding 6 months [37].” Materials and Methods section, line 150-156.

7. Line 134: what does 'status' refer to? the HCV status of sexual partners? Please clarify.

Answer:

Thank for the editor’s suggestion, and we have revised it as “having a sexual partner within the preceding 6 months (assessment options were no sexual partners, one regular sexual partner, no regular sexual partners/less than five partners, and no regular sexual partner/more than five partners) [37], and engaging in other activities involving sexual contact within the preceding 6 months [37]” Materials and Methods section, line 152-155.

We also revised it in the table 1 and 2.

8. Line 142: again, what are the Domains - please describe these.

Answer:

We have revised it as “Secondary outcomes were factors associated with UoHCV and the associations of UoHCV with the mean scores for three domains of HCV knowledge (route of HCV transmission, HCV course and complications, and HCV treatment) among the participants stratified by their HCV serostatus.” Materials and Methods section, line 160-164.

9. Line 152: Please omit the word univariate and use the word bivariate when describing analyses comparing two variables. (univariate describes one variable).

Answer:

Thank for the editor’s valuable suggestion, and we have corrected it as the editor’s suggestion.

10: Line 153Clarify what statistic was used to assess bivariate associations (it is not clear why use binary backwards logistic regression?? ). this sentence should probably read: Binary logistic Binary regression was used to assess associations and calculate odds ratios in bivariate analyses between surveyed factors and UoHCV... in .

Answer:

Thank for the editor’s valuable suggestion, and we have revised it as “Backward stepwise binary logistic regressions were performed to calculate odds ratios and evaluate associations in the bivariate and multivariable analyses between surveyed factors and UoHCV among all the participants and among those in the two HCV serostatus groups. To simultaneously consider the effects of all variables in the multivariable model, we adopted a backward approach.” Materials and Methods section, line 174-178.

11. Line 163: Consider editing this to: To determine HCV-related knowledge and domain scores of knowledge independently associated with UoHCV, we used a multilinear regressions model with backward strategy. Also, please clarify why Backward approach was used.

Answer:

Thank for the editor’s valuable suggestion, and we have revised it as “Finally, to determine the association of UoHCV with the means of the total and domain-specific scores of HCV-related knowledge, we employed a multilinear regression model with a backward approach. β along with 95% confidence intervals were calculated to estimate the effects of UoHCV and directions of all associations. A backward approach was also adopted to enable the effects of all the variables to be simultaneously considered in the multivariable model.” Materials and Methods section, line 188-193. 

RESULTS

1. Lines 176-177: State what the subgroups are in the text.

Answer:

Thank for the editor’s valuable suggestion, and we have revised it as “A total of 844 PLWH were included in the final analysis. They were divided into HCV-seronegative (n = 734) and HCV-seropositive (n = 110) groups. The two groups were further divided into subgroups 1 (unawareness/HCV-seronegative; n = 448), 2 (awareness/HCV-seronegative; n = 286), 3 (unawareness/HCV-seropositive; n = 22), and 4 (awareness/HCV-seropositive; n = 88).” Results section, line 202-207.

2. Table 2: please change the word 'univariable' to Bivariate in the title and in the Table.

Answer:

Thank for the editor’s valuable suggestion, and we have corrected it.

3. Lines 245-247 - the domains noted here should be moved to methods. It is find to reiterate them here as well.

Answer:

Thank for the editor’s valuable suggestion, and we have revised it in the Materials and Methods under the section of “statistical analysis”, as ” We also performed exploratory factor analysis by using principal axis factoring with varimax rotation to investigate the structural domain of the 15 items, and three domains were finally categorized: route of HCV transmission (domain 1), HCV course and complications (domain 2), and HCV treatment (domain 3).” Materials and Methods section, line 181-185.

4. Table 4. Please Name the Domains, not just the number. in the table heading. Can you convert the Betas and 95% CI to adjusted odds ratios.

Answer: 

Thank for the editor’s valuable suggestion, and we have added the name of each domain in Table 4. 

For the association of UoHCV with HCV knowledge scores, the outcome “HCV knowledge” in multivariable model is continuous variable. Therefore, we used multilinear regressions model for analysis, and used “β” (for continuous outcome) to estimate the effect (odds ratios is used for categorical outcome). We also revised it as “β along with 95% confidence intervals were calculated to estimate the effects of UoHCV and directions of all associations.” Materials and Methods section, line 190-191.

DISCUSSION

1. Line 276: Be very sure of this... it may be the first in a study of HIV positive people (but its not the first study to assess discordance in HCV status and knowledge of status.)

Answer:

Thank for the editor’s valuable suggestion, and we have revised it as “To the best of our knowledge, ours is the first study to reveal differences in the prevalence rates of UoHCV among PLWH; these rates were 61.0% in the HCV-seronegative group and 20.0% in the HCV-seropositive group. The prevalence rate in the HCV-seropositive group increased to 33.8% after the exclusion of participants with a history of HCV treatment, all of whom knew their HCV status. After stratification by HCV serostatus, the two groups differed in terms of their sociodemographic characteristics and laboratory variables associated with UoHCV, indicating the need for strategies to be tailored according to HCV serostatus when attempting to reduce UoHCV among PLWH.” Discussion section, line 309-317.

2. Lines 282-283: what does this mean (need for customized strategies). This sentence is too broad - customized to who? or should messages be broadened to other groups. Be more specific.

Answer:

Thank for the editor’s valuable suggestion, and we have revised it as “After stratification by HCV serostatus, the two groups differed in terms of their sociodemographic characteristics and laboratory variables associated with UoHCV, indicating the need for strategies to be tailored according to HCV serostatus when attempting to reduce UoHCV among PLWH.” Discussion section, line 313-317.

3. Line 285 and elsewhere: change IDU to PWID.

Answer:

Thank for the editor’s suggestion, and we have corrected it throughout the manuscript.

4. Line 290: delete 'prevalent among" and change to "increasingly detected"

Answer:

Thank for the editor’s suggestion. We have revised it as “However, since the mid-2000s, sexually transmitted HCV has been increasingly detected among sexually active MSM”. Discussion section, line 323-325.

5. Line 291 and elsewhere: change homosexuals to MSM.

Answer:

Thank for the editor’s suggestion, and we have corrected it throughout the manuscript.

6. Line 294: delete medical, and replace with injection

Answer:

Thank for the editor’s suggestion, and we have revised it as “(e.g., through the use of unsterile injection equipment and contaminated blood products)”. Discussion section, line 328-329.

7. Line 295: delete "or sharing of unsterile equipment" as it is redundant with the above.

Answer:

Thank for the editor’s suggestion, and we have deleted it.

8. Line 302: add "HIV positive" before the word 'individuals'

Answer:

Thank for the editor’s suggestion, and we have added it.

9. Line 306: Add "newly available DAAs" to this sentence.

Answer:

We have revised it as “These efforts include the increase in publicity on HCV worldwide, newly available DAAs, and implementation of measures to ensure the affordability of oral DAA agents in Taiwan.” Discussion section, line 340-342.

10. Lines 316-318: Reframe this sentence. There is risk of reinfection among people treated with HCV who have ongoing exposures. Limiting these is needed to maintain elimination progress.

Answer:

We have revised it as “Reducing the engagement in high-risk behavior is critical for reducing rates of HCV infection. People who have received treatment for HCV who have ongoing exposure to HCV are at risk of reinfection. Limiting such exposure is necessary for making continual progress toward eradication of the disease.” Discussion section, line 353-356.

11. Line 320 - others researchers have not seen behavior change: (i) https://www.sciencedirect.com/science/article/abs/pii/S0376871609002294?via%3Dihub

(ii) https://jech.bmj.com/content/69/8/745

But note that these studies were not conducted in PLWH

Answer

Brief introduction of the two studies:

1. https://www.sciencedirect.com/science/article/abs/pii/S0376871609002294?via%3Dihub:

This prospective study of 112 young PWID who experienced HCV seroconversion during follow-up demonstrated a modest reduction on alcohol and non-injection drug use that is not sustained over time after the young IDU were aware of their HCV seroconversion.

2. https://pubmed.ncbi.nlm.nih.gov/25814695/

This within-cohort matching study enrolled 190 HCV-positive and HCV-negative participnts at baseline. This study demonstrate a 5% per 3-month reduction post-notification in the odds of recent injection drug use (adjusted Odds Ratio: aOR 0.95, 95% CI 0.93–0.96) in HCV-Positive group and a 8% per 3-month reduction post-notification in the odds of Recent syringe borrowing (adjusted Odds Ratio: aOR 0.92, 95% CI 0.87–0.97) in HCV-Negative group.

These two studies provided evidence of reduction of HCV-risk behaviors post-notification of HCV status. We revised it as “Studies have revealed that awareness of HCV infection status reduces engagement in high-risk behaviors among HCV-seropositive PWID [9-12]. However, a significant association was not observed between UoHCV and high-risk behaviors related to HCV infection in the present study, in which the majority (94.8%) of participants had sexually transmitted HIV infection.” Discussion section, line 356-361.

12. Line 336: When you say 'further research' be more engaged. Like what research is needed.

Answer:

We have revised it as “Therefore, to optimize the awareness of the HCV care cascade among HCV-seropositive PLWH [8], further research is necessary to identify effective interventions for modifiable behavioral risk factors and settings during the process of notifying patients of their HCV status to promote long-term reductions in behaviors that involve a high-risk of exposure to HCV [48].” Discussion section, line 379-383.

13. Lines 337-339: This is a very old reference. Many attitudes have changed. Does stigma remain high in Taiwan? Perhaps local reference might be better here to better frame what is needed in Taiwan.

Answer:

Thank for the editor’s valuable suggestion. We have replaced the old reference with a new reference 47“AIDS Patient Care STDS. 2020 Jul;34(7):303-315”, and revised it as “Moreover, interventions to reduce modifiable behavioral risk factors, such as condom distribution, the promotion of abstinence from illicit drugs, and advocacy for safe sex, may attract serious criticism because of the stigma associated with sex and drug use [47].” Discussion section, line 375-379.

14. Lines 341-342: delete part of the sentence that says: ..."particularly among individuals... before 2008"

Answer:

We have deleted it as the editor’s suggestion.

15. Line 343: When saying additional study, should be conducted - like what ? besides condom use?

Answer:

We have rewritten the sentence to reduce the redundancy: Although notifying PWID of their HCV status can effectively reduce their engagement in high-risk behaviors [9-12], effective behavior modification approaches for reducing the sexual transmission of HCV in at-risk populations have yet to be identified. Moreover, interventions to reduce modifiable behavioral risk factors, such as condom distribution, the promotion of abstinence from illicit drugs, and advocacy for safe sex, may attract serious criticism because of the stigma associated with sex and drug use [47]. Therefore, to optimize the awareness of the HCV care cascade among HCV-seropositive PLWH [8], further research is necessary to identify effective interventions for modifiable behavioral risk factors and settings during the process of notifying patients of their HCV status to promote long-term reductions in behaviors that involve a high-risk of exposure to HCV [48]. Discussion section, line 372-383.

16. Line 354-355". What is the medical guideline for HCV testing in HIV positive people in Taiwan? this would be good information to share and make the point that the HCV testing rate is low in this high risk group.

Answer:

Thank for the editor’s valuable suggestion, and have added “According to Taiwan’s guidelines for HIV and acquired immunodeficiency syndrome, HCV-seronegative PLWH should undergo annual screening for HCV antibodies [49].” Discussion section, line 393-394.

17. Lines 357-358" The sentence states there is a negative association: please interpret this for our readers. What does this mean?

Answer:

Thank for the editor’s valuable suggestion, and we have revised it as “However, our data revealed that UoHCV is associated with lower mean scores for total and domain-specific HCV-related knowledge compared with HCV status awareness among PLWH with HCV-seronegative status.” Discussion section, line 396-398.

18. Several places -where Tables are referred to in the Discussion. I think you do not have to tell the reader to go back to the tables and can delete those.

Answer:

We have deleted it throughout the manuscript.

19. Line 376: please make sure to state the population a the end of the sentence ".. in PLWH".

Answer:

We have revised it as “This is the first study to analyze UoHCV among PLWH and identify different determinants associated with UoHCV among PLWH according to their HCV serostatus. The study results also aid in the customization of strategies according to their HCV serostatus for reducing the prevalence of UoHCV among PLWH.” Discussion section, line 414-418.

20: Line 388: Good job on the limitations. I suggest deleting the words "was unavoidable" from this sentence though.

Answer:

If the words “was unavoidable”, this would not be a whole sentence. Therefore, we decided to keep “was unavoidable”, but revised it as “Second, although the trained investigators instructed participants to recall their experiences to answer certain questions, inaccurate recall was unavoidable.” Discussion section, line 429-431.

21. Line 394 change this sentence to read: ".. findings may not be generalizable to all PLWH populations. [i feel that in many places the authors lose sight that this is all done in PLWH - so discussion points need to emphasize that].

Answer:

Thank for the editor’s valuable suggestion. We have revised it as “Finally, the participants in the present study were PLWH, most of whom were MSMs. Therefore, the findings may not be generalizable to all PLWH populations.” Discussion section, line 436-438.

CONCLUSION

1. Line 399" say why the 'era of DAA's is important (eg., the high cure rates and potential to impact population prevalence or lead to elimination...?)

Answer:

We have revised it as “With the current availability of all-oral, interferon-free DAAs, which achieve a curative outcome in more than 90% of patients with HCV and require only 12 weeks of treatment, the eradication of HCV in prevalent areas is now possible [52]; however, our study revealed that the majority of PLWH in Taiwan were unaware of their HCV serostatus.” Conclusion section, line 441-445.

2. Line 405: Delete the end of the sentence starting with 'should' and change to "may be more effective if targeted to PLWH by HCV serostastus. (stay with your data and your population'. (the word personalized is awkward)

Answer:

We have revised it as the editor’s suggestion. “Because the factors associated with UoHCV differed between HCV-seronegative and HCV-seropositive PLWH, strategies for reducing UoHCV may be more effective if they target PLWH according to their HCV serostastus.” Conclusion section, line 449-452.

---

## [Editor Report · Decision Letter 1]

21 Apr 2021

High prevalence of unawareness of HCV infection status among both HCV-seronegative and seropositive people living with human immunodeficiency virus in Taiwan

PONE-D-20-37478R1

Dear Dr. Lu,

We’re pleased to inform you that your manuscript has been judged scientifically suitable for publication and will be formally accepted for publication once it meets all outstanding technical requirements. 

Kind regards,

Kimberly Page, PhD, MPH

Academic Editor

PLOS ONE

Additional Editor Comments (optional):

Thanks to the authors for their considerate and responsive revisions.
---

## [Editor Report · Acceptance letter]

26 Apr 2021

PONE-D-20-37478R1 

High prevalence of unawareness of HCV infection status among both HCV-seronegative and seropositive people living with human immunodeficiency virus in Taiwan 

Dear Dr. Lu:

I'm pleased to inform you that your manuscript has been deemed suitable for publication in PLOS ONE. Congratulations! Your manuscript is now with our production department. 

Kind regards, 

on behalf of

Dr. Kimberly Page 

Academic Editor

PLOS ONE